# Coevolutionary Latent Feature Processes for Continuous-Time User-Item Interactions

**Yichen Wang$^\diamond$, Nan Du$^*$, Rakshit Trivedi$^\diamond$, Le Song$^\diamond$**
$^*$Google Research
$^\diamond$College of Computing, Georgia Institute of Technology
{yichen.wang, rstrivedi}@gatech.edu, dunan@google.com
lsong@cc.gatech.edu

## Abstract

Matching users to the right items at the right time is a fundamental task in recommendation systems. As users interact with different items over time, users' and items' feature may evolve and co-evolve over time. Traditional models based on static latent features or discretizing time into epochs can become ineffective for capturing the fine-grained temporal dynamics in the user-item interactions. We propose a coevolutionary latent feature process model that accurately captures the coevolving nature of users' and items' feature. To learn parameters, we design an efficient convex optimization algorithm with a novel low rank space sharing constraints. Extensive experiments on diverse real-world datasets demonstrate significant improvements in user behavior prediction compared to state-of-the-arts.

## 1   Introduction

Online social platforms and service websites, such as Reddit, Netflix and Amazon, are attracting thousands of users every minute. Effectively recommending the appropriate service items is a fundamentally important task for these online services. By understanding the needs of users and serving them with potentially interesting items, these online platforms can improve the satisfaction of users, and boost the activities or revenue of the sites due to increased user postings, product purchases, virtual transactions, and/or advertisement clicks [30, 9].

As the famous saying goes "You are what you eat and you think what you read", both users' interests and items' semantic features are dynamic and can *evolve* over time [18, 4]. The interactions between users and service items play a critical role in driving the evolution of user interests and item features. For example, for movie streaming services, a long-time fan of comedy watches an interesting science fiction movie one day, and starts to watch more science fiction movies in place of comedies. Likewise, a single movie may also serve different segment of audiences at different times. For example, a movie initially targeted for an older generation may become popular among the younger generation, and the features of this movie need to be redefined.

Another important aspect is that users' interests and items' features can *co-evolve* over time, that is, their evolutions are intertwined and can influence each other. For instance, in online discussion forums, such as Reddit, although a group (item) is initially created for political topics, users with very different interest profiles can join this group (**user** → **item**). Therefore, the participants can shape the actual direction (or features) of the group through their postings and responses. It is not unlikely that this group can eventually become one about education simply because most users here concern about education (**item** → **user**). As the group is evolving towards topics on education, some users may become more attracted to education topics, and to the extent that they even participate in other dedicated groups on education. On the opposite side, some users may gradually gain interests in sports groups, lose interests in political topics and become inactive in this group. Such coevolutionary nature of user-item interactions raises very interesting questions on how to model them elegantly and how to learn them from observed interaction data.

Nowadays, user-item interaction data are archived in increasing temporal resolution and becoming increasingly available. Each individual user-item iteration is typically logged in the database with the precise time-stamp of the interaction, together with additional context of that interaction, such as tag, text, image, audio and video. Furthermore, the user-item interaction data are generated in an *asynchronous* fashion in a sense that any user can interact with any item at any time and there may not be any coordination or synchronization between two interaction events. These types of event data call for new representations, models, learning and inference algorithms.

Despite the temporal and asynchronous nature of such event data, for a long-time, the data has been treated predominantly as a static graph, and fixed latent features have been assigned to each user and item [21, 5, 2, 10, 29, 30, 25]. In more sophisticated methods, the time is divided into epochs, and static latent feature models are applied to each epoch to capture some temporal aspects of the data [18, 17, 28, 6, 13, 4, 20, 17, 28, 12, 15, 24, 23]. For such epoch-based methods, it is not clear how to choose the epoch length parameter due to the asynchronous nature of the user-item interactions. First, different users may have very different time-scale when they interact with those service items, making it very difficult to choose a unified epoch length. Second, it is not easy for the learned model to answer fine-grained time-sensitive queries such as when a user will come back for a particular service item. It can only make such predictions down to the resolution of the chosen epoch length. Most recently, [9] proposed an efficient low-rank point process model for time-sensitive recommendations from recurrent user activities. However, it still fails to capture the heterogeneous coevolutionary properties of user-item interactions with much more limited model flexibility. Furthermore, it is difficult for this approach to incorporate observed context features.

In this paper, we propose a coevolutionary latent feature process for continuous-time user-item interactions, which is designed specifically to take into account the asynchronous nature of event data, and the co-evolution nature of users' and items' latent features. Our model assigns an evolving latent feature process for each user and item, and the co-evolution of these latent feature processes is considered using two parallel components:

- (**Item → User**) A user's latent feature is determined by the latent features of the items he interacted with. Furthermore, the contributions of these items' features are temporally discounted by an exponential decaying kernel function, which we call the Hawkes [14] feature averaging process.
- (**User → Item**) Conversely, an item's latent features are determined by the latent features of the users who interact with the item. Similarly, the contribution of these users' features is also modeled as a Hawkes feature averaging process.

Besides the two sets of intertwined latent feature processes, our model can also take into account the presence of potentially high dimensional observed context features and links the latent features to the observed context features using a low dimensional projection. Despite the sophistication of our model, we show that the model parameter estimation, a seemingly non-convex problem, can be transformed into a convex optimization problem, which can be efficiently solved by the latest conditional gradient-like algorithm. Finally, the coevolutionary latent feature processes can be used for down-streaming inference tasks such as the next-item and the return-time prediction. We evaluate our method over a variety of datasets, verifying that our method can lead to significant improvements in user behavior prediction compared to the state-of-the-arts.

## 2 Background on Temporal Point Processes

This section provides necessary concepts of the temporal point process [7]. It is a random process whose realization consists of a list of events localized in time, $\{t_i\}$ with $t_i \in \mathbb{R}^+$. Equivalently, a given temporal point process can be represented as a counting process, $N(t)$, which records the number of events before time $t$. An important way to characterize temporal point processes is via the conditional intensity function $\lambda(t)$, a stochastic model for the time of the next event given all the previous events. Formally, $\lambda(t)\mathrm{d}t$ is the conditional probability of observing an event in a small window $[t, t+\mathrm{d}t)$ given the history $\mathcal{T}(t)$ up to $t$, *i.e.*, $\lambda(t)\mathrm{d}t := \mathbb{P}\{\text{event in } [t, t+\mathrm{d}t)|\mathcal{T}(t)\} = \mathbb{E}[\mathrm{d}N(t)|\mathcal{T}(t)]$, where one typically assumes that only one event can happen in a small window of size $\mathrm{d}t$, *i.e.*, $\mathrm{d}N(t) \in \{0, 1\}$.

The function form of the intensity is often designed to capture the phenomena of interests. One commonly used form is the Hawkes process [14, 11, 27, 26], whose intensity models the excitation between events, *i.e.*, $\lambda(t) = \mu + \alpha \sum_{t_i \in \mathcal{T}(t)} \kappa_\omega(t - t_i)$, where $\kappa_\omega(t) := \exp(-\omega t)$ is an exponential triggering kernel, $\mu \geqslant 0$ is a baseline intensity independent of the history. Here, the occurrence of each historical event increases the intensity by a certain amount determined by the kernel $\kappa_\omega$ and the weight $\alpha \geqslant 0$, making the intensity history dependent and a stochastic process by itself. From

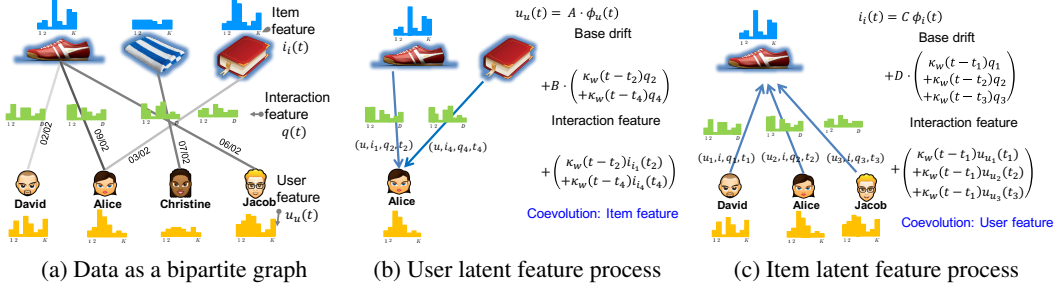

|(a) Data as a bipartite graph|(b) User latent feature process|(c) Item latent feature process|

Figure 1: Model illustration. (a) User-item interaction events data. Each edge contains user, item, time, and interaction feature. (b) Alice's latent feature consists of three components: the drift of baseline feature, the time-weighted average of interaction feature, and the weighted average of item feature. (c) The symmetric item latent feature process. $\boldsymbol{A}, \boldsymbol{B}, \boldsymbol{C}, \boldsymbol{D}$ are embedding matrices from high dimension feature space to latent space. $\kappa_\omega(t) = \exp(-\omega t)$ is an exponential decaying kernel.

the survival analysis theory [1], given the history $\mathcal{T} = \{t_1, \dots, t_n\}$, for any $t > t_n$, we characterize the conditional probability that no event happens during $[t_n, t)$ as $S(t|\mathcal{T}) = \exp\left(-\int_{t_n}^t \lambda(\tau)\,\mathrm{d}\tau\right)$. Moreover, the conditional density that an event occurs at time $t$ is $f(t|\mathcal{T}) = \lambda(t)\,S(t|\mathcal{T})$.

# 3 Coevolutionary Latent Feature Processes

In this section, we present the framework to model the temporal dynamics of user-item interactions. We first explicitly capture the co-evolving nature of users' and items' latent features. Then, based on the compatibility between a user' and item's latent feature, we model the user-item interaction by a temporal point process and parametrize the intensity function by the feature compatibility.

## 3.1 Event Representation

Given $m$ users and $n$ items, the input consists of all users' history events: $\mathcal{T} = \{e_k\}$, where $e_k = (u_k, i_k, t_k, \boldsymbol{q}_k)$ means that user $u_k$ interacts with item $i_k$ at time $t_k$ and generates an interaction feature vector $\boldsymbol{q}_k \in \mathbb{R}^D$. For instance, the interaction feature can be a textual message delivered from the user to the chatting-group in Reddit or a review of the business in Yelp. It can also be unobservable if the data only contains the temporal information.

## 3.2 Latent Feature Processes

We associate a latent feature vector $\boldsymbol{u}_u(t) \in \mathbb{R}^K$ with a user $u$ and $\boldsymbol{i}_i(t) \in \mathbb{R}^K$ with an item $i$. These features represent the subtle properties which cannot be directly observed, such as the interests of a user and the semantic topics of an item. Specifically, we model $\boldsymbol{u}_u(t)$ and $\boldsymbol{i}_i(t)$ as follows:

*User latent feature process.* For each user $u$, we formulate $\boldsymbol{u}_u(t)$ as:

$$\boldsymbol{u}_u(t) = \boldsymbol{A}\underbrace{\boldsymbol{\phi}_u(t)}_{\text{base drift}} + \boldsymbol{B}\underbrace{\sum_{\{e_k|u_k=u, t_k<t\}} \kappa_\omega(t-t_k)\boldsymbol{q}_k}_{\text{Hawkes interaction feature averaging}} + \underbrace{\sum_{\{e_k|u_k=u, t_k<t\}} \kappa_\omega(t-t_k)\boldsymbol{i}_{i_k}(t_k)}_{\text{co-evolution: Hawkes item feature averaging}}, \quad (1)$$

*Item latent feature process.* For each item $i$, we specify $\boldsymbol{i}_i(t)$ as:

$$\boldsymbol{i}_i(t) = \boldsymbol{C}\underbrace{\boldsymbol{\phi}_i(t)}_{\text{base drift}} + \boldsymbol{D}\underbrace{\sum_{\{e_k|i_k=i, t_k<t\}} \kappa_\omega(t-t_k)\boldsymbol{q}_k}_{\text{Hawkes interaction feature averaging}} + \underbrace{\sum_{\{e_k|i_k=i, t_k<t\}} \kappa_\omega(t-t_k)\boldsymbol{u}_{u_k}(t_k)}_{\text{co-evolution: Hawkes user feature averaging}}, \quad (2)$$

where $\boldsymbol{A}, \boldsymbol{B}, \boldsymbol{C}, \boldsymbol{D} \in \mathbb{R}^{K \times D}$ are the embedding matrices mapping from the explicit high-dimensional feature space into the low-rank latent feature space. Figure 1 highlights the basic setting of our model. Next we discuss the rationale of each term in detail.

**Drift of base features**. $\boldsymbol{\phi}_u(t) \in \mathbb{R}^D$ and $\boldsymbol{\phi}_i(t) \in \mathbb{R}^D$ are the explicitly observed properties of user $u$ and item $i$, which allows the basic features of users (*e.g.*, a user's self-crafted interests) and items (*e.g.*, textual categories and descriptions) to smoothly drift through time. Such changes of basic features normally are caused by external influences. One can parametrize $\boldsymbol{\phi}_u(t)$ and $\boldsymbol{\phi}_i(t)$ in many different ways, *e.g.*, the exponential decaying basis to interpolate these features observed at different times.

**Evolution with interaction feature**. Users' and items' features can evolve and be influenced by the characteristics of their interactions. For instance, the genre changes of movies indicate the changing tastes of users. The theme of a chatting-group can be easily shifted to certain topics of the involved discussions. In consequence, this term captures the cumulative influence of the past interaction features to the changes of the latent user (item) features. The triggering kernel $\kappa_\omega(t - t_k)$ associated with each past interaction at $t_k$ quantifies how such influence can change through time. Its parametrization depends on the phenomena of interest. Without loss of generality, we choose the exponential kernel $\kappa_\omega(t) = \exp(-\omega t)$ to reduce the influence of each past event. In other words, only the most recent interaction events will have bigger influences. Finally, the embedding $\boldsymbol{B}, \boldsymbol{D}$ map the observable high dimension interaction feature to the latent space.

**Coevolution with Hawkes feature averaging processes**. Users' and items' latent features can mutually influence each other. This term captures the two parallel processes:

- *Item → User*. A user's latent feature is determined by the latent features of the items he interacted with. At each time $t_k$, the latent item feature is $\boldsymbol{i}_{i_k}(t_k)$. Furthermore, the contributions of these items' features are temporally discounted by a kernel function $\kappa_\omega(t)$, which we call the Hawkes feature averaging process. The name comes from the fact that Hawkes process captures the temporal influence of history events in its intensity function. In our model, we capture both the temporal influence and feature of each history item as a latent process.
- *User → Item*. Conversely, an item's latent features are determined by the latent features of all the users who interact with the item. At each time $t_k$, the latent feature is $\boldsymbol{u}_{u_k}(t_k)$. Similarly, the contribution of these users' features is also modeled as a Hawkes feature averaging process.

Note that to compute the third co-evolution term, we need to keep track of the user's and item's latent features after each interaction event, *i.e.*, at $t_k$, we need to compute $\boldsymbol{u}_{u_k}(t_k)$ and $\boldsymbol{i}_{i_k}(t_k)$ in (1) and (2), respectively. Set $\mathbb{I}(\cdot)$ to be the indicator function, we can show by induction that

$$\boldsymbol{u}_{u_k}(t_k) = \boldsymbol{A}\Big[\sum_{j=1}^{k} \mathbb{I}[u_j = u_k]\kappa_\omega(t_k - t_j)\boldsymbol{\phi}_{u_j}(t_j)\Big] + \boldsymbol{B}\Big[\sum_{j=1}^{k} \mathbb{I}[u_j = u_k]\kappa_\omega(t_k - t_j)\boldsymbol{q}_j\Big]$$

$$+ \boldsymbol{C}\Big[\sum_{j=1}^{k-1} \mathbb{I}[u_j = u_k]\kappa_\omega(t_k - t_j)\boldsymbol{\phi}_{i_j}(t_j)\Big] + \boldsymbol{D}\Big[\sum_{j=1}^{k-1} \mathbb{I}[u_j = u_k]\kappa_\omega(t_k - t_j)\boldsymbol{q}_j\Big]$$

$$\boldsymbol{i}_{i_k}(t_k) = \boldsymbol{C}\Big[\sum_{j=1}^{k} \mathbb{I}[i_j = i_k]\kappa_\omega(t_k - t_j)\boldsymbol{\phi}_{i_j}(t_j)\Big] + \boldsymbol{D}\Big[\sum_{j=1}^{k} \mathbb{I}[i_j = i_k]\kappa_\omega(t_k - t_j)\boldsymbol{q}_j\Big]$$

$$+ \boldsymbol{A}\Big[\sum_{j=1}^{k-1} \mathbb{I}[i_j = i_k]\kappa_\omega(t_k - t_j)\boldsymbol{\phi}_{u_j}(t_j)\Big] + \boldsymbol{B}\Big[\sum_{j=1}^{k-1} \mathbb{I}[i_j = i_k]\kappa_\omega(t_k - t_j)\boldsymbol{q}_j\Big]$$

In summary, we have incorporated both of the exogenous and endogenous influences into a single model. First, each process evolves according to the respective exogenous base temporal user (item) features $\phi_u(t)$ ($\phi_i(t)$). Second, the two processes also inter-depend on each other due to the endogenous influences from the interaction features and the entangled latent features. We present our model in the most general form and the specific choices of $\boldsymbol{u}_u(t), \boldsymbol{i}_i(t)$ are dependent on applications. For example, if no interaction feature is observed, we drop the second term in (1) and (2).

### 3.3 User-Item Interactions as Temporal Point Processes

For each user, we model the recurrent occurrences of user $u$'s interaction with all items as a multi-dimensional temporal point process. In particular, the intensity in the $i$-th dimension (item $i$) is:

$$\lambda^{u,i}(t) = \underbrace{\eta^{u,i}}_{\text{long-term preference}} + \underbrace{\boldsymbol{u}_u(t)^\top \boldsymbol{i}_i(t)}_{\text{short-term preference}}, \tag{3}$$

where $\boldsymbol{\eta} = (\eta^{u,i})$ is a baseline preference matrix. The rationale of this formulation is threefold. First, instead of discretizing the time, we explicitly model the timing of each event occurrence as a continuous random variable, which naturally captures the heterogeneity of the temporal interactions between users and items. Second, the base intensity $\eta^{u,i}$ represents the long-term preference of user $u$ to item $i$, independent of the history. Third, the tendency for user $u$ to interact with item $i$ at time $t$ depends on the compatibility of their instantaneous latent features. Such compatibility is evaluated through the inner product of their time-varying latent features.

Our model inherits the merits from classic content filtering, collaborative filtering, and the most recent temporal models. For the cold-start users having few interactions with the items, the model adaptively utilizes the purely observed user (item) base properties and interaction features to adjust its predictions, which incorporates the key idea of feature-based algorithms. When the observed

features are missing and non-informative, the model makes use of the user-item interaction patterns to make predictions, which is the strength of collaborative filtering algorithms. However, being different from the conventional matrix-factorization models, the latent user and item features in our model are entangled and able to co-evolve over time. Finally, the general temporal point process ingredient of the model makes it possible to capture the dynamic preferences of users to items and their recurrent interactions, which is more flexible and expressive.

## 4  Parameter Estimation

In this section, we propose an efficient framework to learn the parameters. A key challenge is that the objective function is non-convex in the parameters. However, we reformulate it as a convex optimization by creating new parameters. Finally, we present the generalized conditional gradient algorithm to efficiently solve the objective function.

Given a collection of events $\mathcal{T}$ recorded within a time window $[0, T)$, we estimate the parameters using maximum likelihood estimation of all events. The joint negative log-likelihood [1] is:

$$\ell = -\sum_{e_k} \log\left(\lambda^{u_k, i_k}(t_k)\right) + \sum_{u=1}^{m}\sum_{i=1}^{n}\int_0^T \lambda^{u,i}(\tau)\,d\tau \tag{4}$$

The objective function is non-convex in variables $\{A, B, C, D\}$ due to the inner product term in (3). To learn these parameters, one way is to fix the matrix rank and update each matrix using gradient based methods. However, it is easily trapped in local optima and one needs to tune the rank for the best performance. However, with the observation that the product of two low rank matrices yields a low rank matrix, we will optimize over the new matrices and obtain a convex objective function.

### 4.1  Convex Objective Function

We will create new parameters such that the intensity function is convex. Since $\boldsymbol{u}_u(t)$ contains the averaging of $\boldsymbol{i}_{i_k}(t_k)$ in (1), $C, D$ will appear in $\boldsymbol{u}_u(t)$. Similarly, $A, B$ will appear in $\boldsymbol{i}_i(t)$. Hence these matrices $\mathcal{X} = \left\{A^\top A, B^\top B, C^\top C, D^\top D, A^\top B, A^\top C, A^\top D, B^\top C, B^\top D, C^\top D\right\}$ will appear in (3) after expansion, due to the inner product $\boldsymbol{i}_i(t)^\top \boldsymbol{u}_u(t)$. For each matrix product in $\mathcal{X}$, we denote it as a new variable $X_i$ and optimize the objective function over the these variables. We denote the corresponding coefficient of $X_i$ as $x_i(t)$, which can be exactly computed. Denote $\boldsymbol{\Lambda}(t) = (\lambda^{u,i}(t))$, we can rewrite the intensity in (3) in the matrix form as:

$$\boldsymbol{\Lambda}(t) = \boldsymbol{\eta} + \sum_{i=1}^{10} x_i(t)X_i \tag{5}$$

The intensity is convex in each new variable $X_i$, hence the objective function. We will optimize over the new set of variables $\mathcal{X}$ subject to the constraints that i) some of them share the same low rank space, $e.g.$, $A^\top$ is shared as the column space in $\left\{A^\top A, A^\top B, A^\top C, A^\top D\right\}$ and ii) new variables are low rank (the product of low rank matrices is low rank). Next, we show how to incorporate the space sharing constraint for general objective function with an efficient algorithm.

First, we create a $symmetric$ block matrix $X \in \mathbb{R}^{4D \times 4D}$ and place each $X_i$ as follows:

$$X = \begin{pmatrix} X_1 & X_2 & X_3 & X_4 \\ X_2^\top & X_5 & X_6 & X_7 \\ X_3^\top & X_6^\top & X_8 & X_9 \\ X_4^\top & X_7^\top & X_9^\top & X_{10} \end{pmatrix} = \begin{pmatrix} A^\top A & A^\top B & A^\top C & A^\top D \\ B^\top A & B^\top B & B^\top C & B^\top D \\ C^\top A & C^\top B & C^\top C & C^\top D \\ D^\top A & D^\top B & D^\top C & D^\top D \end{pmatrix} \tag{6}$$

Intuitively, minimizing the nuclear norm of $X$ ensures all the low rank space sharing constraints. First, nuclear norm $\|\cdot\|_*$ is a summation of all singular values, and is commonly used as a convex surrogate for the matrix rank function [22], hence minimizing $\|X\|_*$ ensures it to be low rank and gives the unique low rank factorization of $X$. Second, since $X_1, X_2, X_3, X_4$ are in the same row and share $A^\top$, the space sharing constraints are naturally satisfied.

Finally, since it is typically believed that users' long-time preference to items can be categorized into a limited number of prototypical types, we set $\boldsymbol{\eta}$ to be low rank. Hence the objective is:

$$\min_{\boldsymbol{\eta} \geqslant 0, X \geqslant 0} \ell(X, \boldsymbol{\eta}) + \alpha\|\boldsymbol{\eta}\|_* + \beta\|X\|_* + \gamma\|X - X^\top\|_F^2 \tag{7}$$

where $\ell$ is defined in (4) and $\|\cdot\|_F$ is the Frobenius norm and the associated constraint ensures $X$ to be symmetric. $\{\alpha, \beta, \gamma\}$ control the trade-off between the constraints. After obtaining $X$, one can directly apply (5) to compute the intensity and make predictions.

## 4.2 Generalized Conditional Gradient Algorithm

We use the latest generalized conditional gradient algorithm [9] to solve the optimization problem (7). We provide details in the appendix. It has an alternating updates scheme and efficiently handles the nonnegative constraint using the proximal gradient descent and the the nuclear norm constraint using conditional gradient descent. It is guaranteed to converge in $O(\frac{1}{t} + \frac{1}{t^2})$, where $t$ is the number of iterations. For both the proximal and the conditional gradient parts, the algorithm achieves the corresponding *optimal* convergence rates. If there is no nuclear norm constraint, the results recover the well-known optimal $O(\frac{1}{t^2})$ rate achieved by proximal gradient method for smooth convex optimization. If there is no nonnegative constraints, the results recover the well-known $O(\frac{1}{t})$ rate attained by conditional gradient method for smooth convex minimization. Moreover, the per-iteration complexity is linear in the total number of events with $O(mnk)$, where $m$ is the number of users, $n$ is the number of items and $k$ is the number of events per user-item pair.

# 5 Experiments

We evaluate our framework, COEVOLVE, on synthetic and real-world datasets. We use all the events up to time $T \cdot p$ as the training data, and the rest as testing data, where $T$ is the length of the observation window. We tune hyper-parameters and the latent rank of other baselines using 10-fold cross validation with grid search. We vary the proportion $p \in \{0.7, 0.72, 0.74, 0.76, 0.78\}$ and report the averaged results over five runs on two tasks:

(a) **Item recommendation:** for each user $u$, at every testing time $t$, we compute the survival probability $S^{u,i}(t) = \exp\left(-\int_{t_n^{u,i}}^{t} \lambda^{u,i}(\tau)\mathrm{d}\tau\right)$ of each item $i$ up to time $t$, where $t_n^{u,i}$ is the last training event time of $(u,i)$. We then rank all the items in the ascending order of $S^{u,i}(t)$ to produce a recommendation list. Ideally, the item associated with the testing time $t$ should rank one, hence smaller value indicates better predictive performance. We repeat the evaluation on each testing moment and report the Mean Average Rank (MAR) of the respective testing items across all users.

(b) **Time prediction:** we predict the time when a testing event will occur between a given user-item pair $(u,i)$ by calculating the density of the next event time as $f(t) = \lambda^{u,i}(t)S^{u,i}(t)$. With the density, we compute the expected time of next event by sampling future events as in [9]. We report the Mean Absolute Error (MAE) between the predicted and true time. Furthermore, we also report the relative percentage of the prediction error with respect to the entire testing time window.

## 5.1 Competitors

**TimeSVD++** is the classic matrix factorization method [18]. The latent factors of users and items are designed as decay functions of time and also linked to each other based on time. **FIP** is a static low rank latent factor model to uncover the compatibility between user and item features [29]. TSVD++ and FIP are only designed for data with explicit ratings. We convert the series of user-item interaction events into an explicit rating using the frequency of a user's item consumptions [3]. **STIC** fits a semi-hidden markov model to each observed user-item pair [16] and is only designed for time prediction. **PoissonTensor** uses Poisson regression as the loss function [6] and has been shown to outperform factorization methods based on squared loss [17, 28] on recommendation tasks. There are two choices of reporting performance: i) use the parameters fitted only in the last time interval and ii) use the average parameters over all intervals. We report the best performance between these two choices. **LowRankHawkes** is a Hawkes process based model and it assumes user-item interactions are independent [9].

## 5.2 Experiments on Synthetic Data

We simulate 1,000 users and 1,000 items. For each user, we further generate 10,000 events by Ogata's thinning algorithm [19]. We compute the MAE by comparing estimated $\boldsymbol{\eta}$, $\boldsymbol{X}$ with the ground-truth. The baseline drift feature is set to be constant. Figure 2 (a) shows that it only requires a few hundred iterations to descend to a decent error, and (b) indicates that it only requires a modest number of events to achieve a good estimation. Finally, (c) demonstrates that our method scales linearly as the total number of training events grows.

Figure 2 (d-f) show that COEVOLVE achieves the best predictive performance. Because POISSON-TENSOR applies an extra time dimension and fits each time interval as a Poisson regression, it outperforms TIMESVD++ by capturing the fine-grained temporal dynamics. Finally, our method automatically adapts different contributions of each past item factors to better capture the users' current latent features, hence it can achieve the best prediction performance overall.

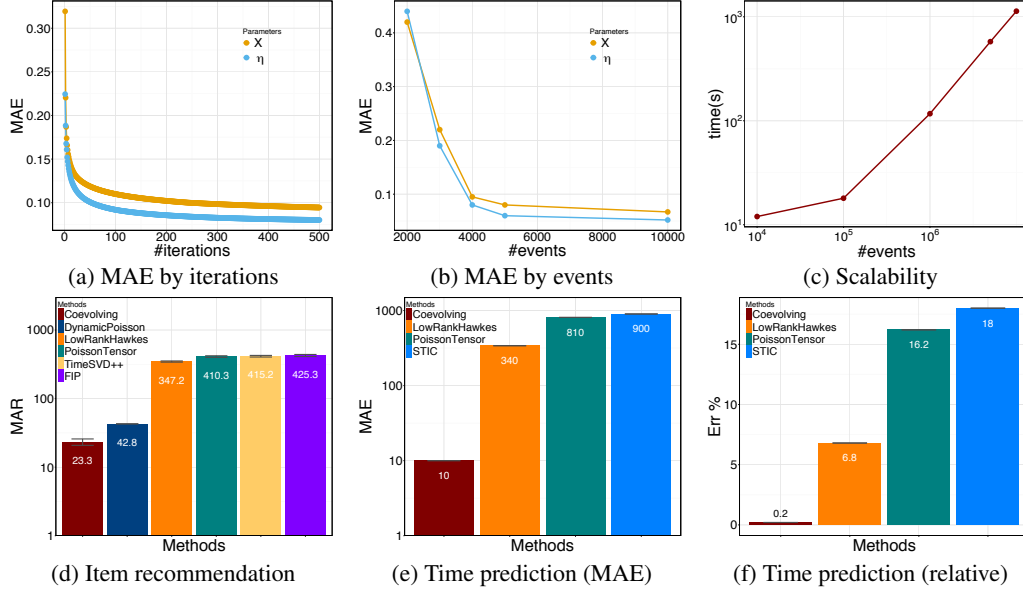

Figure 2: Estimation error (a) vs. #iterations and (b) vs. #events per user; (c) scalability vs. #events per user; (d) average rank of the recommended items; (e) and (f) time prediction error.

## 5.3 Experiments on Real-World Data

**Datasets.** Our datasets are obtained from three different domains from the TV streaming services (IPTV), the commercial review website (Yelp) and the online media services (Reddit). **IPTV** contains 7,100 users' watching history of 436 TV programs in 11 months, with 2,392,010 events, and 1,420 movie features, including 1,073 actors, 312 directors, 22 genres, 8 countries and 5 years. **Yelp** is available from Yelp Dataset challenge Round 7. It contains reviews for various businesses from October, 2004 to December, 2015. We filter users with more than 100 posts and it contains 100 users and 17,213 businesses with around 35,093 reviews. **Reddit** contains the discussions events in January 2014. Furthermore, we randomly selected 1,000 users and collect 1,403 groups that these users have discussion in, with a total of 10,000 discussion events. For item base feature, IPTV has movie feature, Yelp has business description, and Reddit does not have it. In experiments we fix the baseline features. There is no base feature for user. For interaction feature, Reddit and Yelp have reviews in bag-of-words, and no such feature in IPTV.

Figure 3 shows the predictive performance. For time prediction, COEVOLVE outperforms the baselines *significantly*, since we explicitly reason and model the effect that past consumption behaviors change users' interests and items' features. In particular, compared with LOWRANKHAWKES, our model captures the interactions of each user-item pair with a multi-dimensional temporal point processes. It is more expressive than the respective one-dimensional Hawkes process used by LOWRANKHAWKES, which ignores the mutual influence among items. Furthermore, since the unit time is hour, the improvement over the state-of-art on IPTV is around two weeks and on Reddit is around two days. Hence our method significantly helps online services make better demand predictions.

For item recommendation, COEVOLVE also achieves competitive performance comparable with LOWRANKHAWKES on IPTV and Reddit. The reason behind the phenomena is that one needs to compute the rank of the intensity function for the item prediction task, and the value of intensity function for time prediction. LOWRANKHAWKES might be good at differentiating the rank of intensity better than COEVOLVE. However, it may not be able to learn the actual value of the intensity accurately. Hence our method has the order of magnitude improvement in the time prediction task.

In addition to the superb predictive performance, COEVOLVE also learns the time-varying latent features of users and items. Figure 4 (a) shows that the user is initially interested in TV programs of adventures, but then the interest changes to Sitcom, Family and Comedy and finally switches to the Romance TV programs. Figure 4 (b) shows that Facebook and Apple are the two hot topics in the month of January 2014. The discussions about Apple suddenly increased on 01/21/2014, which

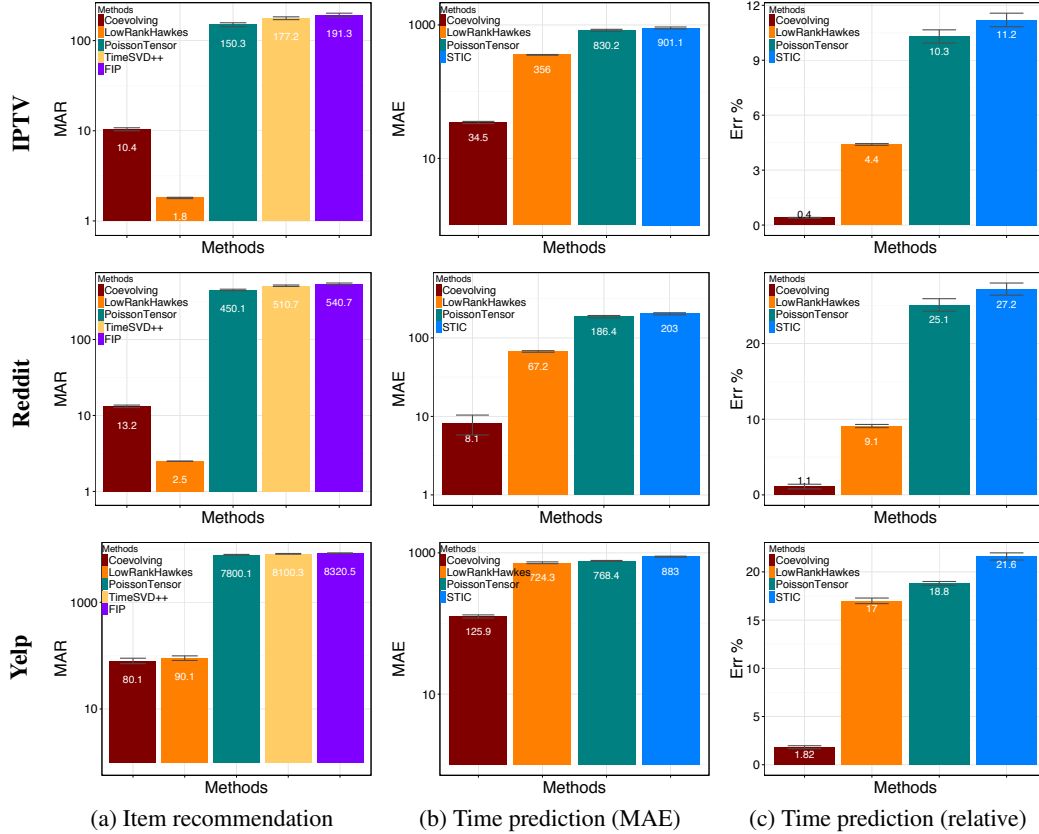

(a) Item recommendation     (b) Time prediction (MAE)     (c) Time prediction (relative)

Figure 3: Prediction results on IPTV, Reddit and Yelp. Results are averaged over five runs with different portions of training data and error bar represents the variance.

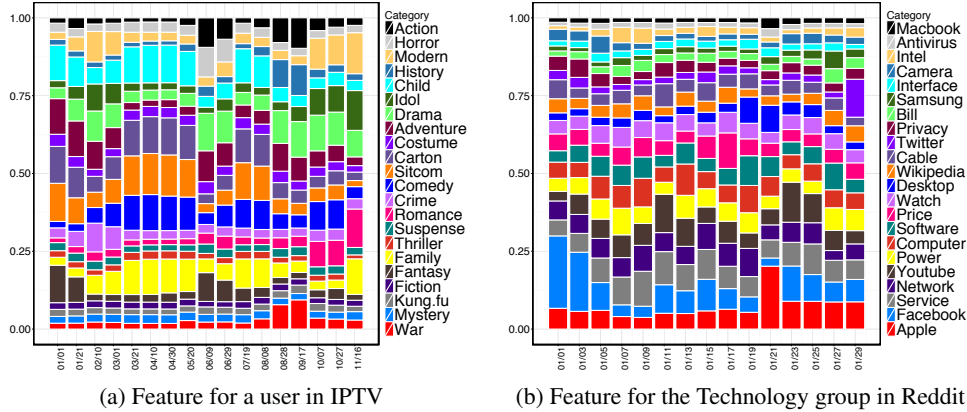

(a) Feature for a user in IPTV     (b) Feature for the Technology group in Reddit

Figure 4: Learned time-varying features of a user in IPTV and a group in Reddit.

can be traced to the news that Apple won lawsuit against Samsung[1]. It further demonstrates that our model can better explain and capture the user behavior in the real world.

# 6 Conclusion

We have proposed an efficient framework for modeling the co-evolution nature of users' and items' latent features. Empirical evaluations on large synthetic and real-world datasets demonstrate its scalability and superior predictive performance. Future work includes extending it to other applications such as modeling dynamics of social groups, and understanding peoples' behaviors on Q&A sites.

**Acknowledge.** This project was supported in part by NSF/NIH BIGDATA 1R01GM108341, ONR N00014-15-1-2340, NSF IIS-1218749, and NSF CAREER IIS-1350983.

## Footnotes

[1] http://techcrunch.com/2014/01/22/apple-wins-big-against-samsung-in-court/

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
