[Supplementary Material · paper_2267_supplementary.pdf]

---

**Algorithm 1** COEVOLUTIONARY LATENT FEATURE PROCESSES

---
1: **Input:** Events $\mathcal{T}$, learning rate $\xi$. **Output:** $\boldsymbol{\eta}, \boldsymbol{X}$
2: Choose to initialize $\boldsymbol{\eta}^0, \boldsymbol{X}^0, \boldsymbol{Z}_1^0, \boldsymbol{Z}_2^0$
3: **for** $k = 1$ **to** *MaxIter* **do**
4:      Compute $\boldsymbol{X}^k = \left(\boldsymbol{X}^{k-1} - \xi \nabla_{\boldsymbol{X}} f(\boldsymbol{X}^{k-1}, \boldsymbol{\eta}^{k-1}, \boldsymbol{Z}_1^{k-1}, \boldsymbol{Z}_2^{k-1})\right)_+$
5:      Compute $\boldsymbol{\eta}^k = \left(\boldsymbol{\eta}^{k-1} - \xi \nabla_{\boldsymbol{\eta}} f(\boldsymbol{X}^{k-1}, \boldsymbol{\eta}^{k-1}, \boldsymbol{Z}_1^{k-1}, \boldsymbol{Z}_2^{k-1})\right)_+$
6:      Find $(\boldsymbol{u}_1, \boldsymbol{v}_1)$ as top singular vector pairs of $-\nabla_{\boldsymbol{Z}_1} f(\boldsymbol{X}^k, \boldsymbol{\eta}^k, \boldsymbol{Z}_1^{k-1}, \boldsymbol{Z}_2^{k-1})$
7:      Find $(\boldsymbol{u}_2, \boldsymbol{v}_2)$ as top singular vector pairs of $-\nabla_{\boldsymbol{Z}_2} f(\boldsymbol{X}^k, \boldsymbol{\eta}^k, \boldsymbol{Z}_1^{k-1}, \boldsymbol{Z}_2^{k-1})$
8:      Set $\delta_k = \frac{2}{k+1}$ and find $\theta_k^i$ by solving $\theta_k^i = \operatorname{argmin}_{\theta \geqslant 0} h^i(\theta_k^i)$ for $i \in \{1, 2\}$.
9:      $\boldsymbol{Z}_1^k = (1 - \delta_k)\boldsymbol{Z}_1^{k-1} + \delta_k \theta_k^1 \boldsymbol{u}_1 \boldsymbol{v}_1^\top$, $\boldsymbol{Z}_2^k = (1 - \delta_k)\boldsymbol{Z}_2^{k-1} + \delta_k \theta_k^2 \boldsymbol{u}_2 \boldsymbol{v}_2^\top$
10: **end for**

---

# A  Generalized Conditional Gradient Algorithm

In this section, we provide details on the latest generalized conditional gradient descent algorithm proposed in [9]. We first provide an alternative formulation of the objective function, and then present the general algorithm.

## A.1  Alternative Formulation

Directly solving the objective (7) is difficult since the nonnegative constraints are entangled with the non-smooth nuclear norm penalty. To address this challenge, we use a simple penalty method. Specifically, given $\rho > 0$, we arrive at the next formulation (8) by introducing two auxiliary variables $\boldsymbol{Z}_1$ and $\boldsymbol{Z}_2$ with some penalty function, such as the squared Frobenius norm.

$$\min_{\boldsymbol{\eta} \geqslant 0, \boldsymbol{X} \geqslant 0, \boldsymbol{Z}_1, \boldsymbol{Z}_2} \ell(\boldsymbol{\eta}, \boldsymbol{X}) + \gamma \|\boldsymbol{X} - \boldsymbol{X}^\top\|_F^2 + \alpha \|\boldsymbol{Z}_1\|_* + \beta \|\boldsymbol{Z}_2\|_* + \rho \|\boldsymbol{\eta} - \boldsymbol{Z}_1\|_F^2 + \rho \|\boldsymbol{X} - \boldsymbol{Z}_2\|_F^2$$
(8)

The new formulation (8) allows us to handle the non-negativity constraints and nuclear norm regularization terms separately.

## A.2  Alternating Updates between Proximal Graident and Conditional Gradient

Now, we present Algorithm 1 that can solve (8) efficiently. For notation simplicity, we first set

$$f(\boldsymbol{\eta}, \boldsymbol{X}, \boldsymbol{Z}_1, \boldsymbol{Z}_2) = \ell(\boldsymbol{\eta}, \boldsymbol{X}) + \gamma \|\boldsymbol{X} - \boldsymbol{X}^\top\|_F^2 + \rho \|\boldsymbol{\eta} - \boldsymbol{Z}_1\|_F^2 + \rho \|\boldsymbol{X} - \boldsymbol{Z}_2\|_F^2$$

At each iteration, we apply cheap projection gradient for block $\{\boldsymbol{\eta}, \boldsymbol{X}\}$ and cheap linear minimization for block $\{\boldsymbol{Z}_1, \boldsymbol{Z}_2\}$. Specifically, the algorithm consists of two main alternating subroutines:

**Proximal Gradient.** When updating $\{\boldsymbol{\eta}, \boldsymbol{X}\}$, we directly compute the associated proximal operator, which in our case, reduces to the simple projection as follows,

$$\boldsymbol{X}^k = \left(\boldsymbol{X}^{k-1} - \xi \nabla_{\boldsymbol{X}} f(\boldsymbol{X}^{k-1}, \boldsymbol{\eta}^{k-1}, \boldsymbol{Z}_1^{k-1}, \boldsymbol{Z}_2^{k-1})\right)_+$$

$$\boldsymbol{\eta}^k = \left(\boldsymbol{\eta}^{k-1} - \xi \nabla_{\boldsymbol{\eta}} f(\boldsymbol{X}^{k-1}, \boldsymbol{\eta}^{k-1}, \boldsymbol{Z}_1^{k-1}, \boldsymbol{Z}_2^{k-1})\right)_+$$

where $(\cdot)_+$ simply sets the negative coordinates to zero.

**Conditional Gradient.** When updating $\{\boldsymbol{Z}_1, \boldsymbol{Z}_2\}$, we use the conditional gradient algorithm that successively linearizes $f$ and finds a descent direction by solving:

$$\boldsymbol{Y}_1^k = \operatorname*{argmin}_{\|\boldsymbol{Y}\|_* \leqslant 1} \left\langle \boldsymbol{Y}, \nabla_{\boldsymbol{Z}_1} f(\boldsymbol{X}^k, \boldsymbol{\eta}^k, \boldsymbol{Z}_1^{k-1}, \boldsymbol{Z}_2^{k-1}) \right\rangle$$
(9)

and then takes the convex combination $\boldsymbol{Z}_1^k = (1 - \delta_k)\boldsymbol{Z}_1^{k-1} + \delta_k \theta_k \boldsymbol{Y}_1^k$ with a suitable step size $\eta_k$ and scaling factor $\theta_k$. The minimizer of (9) is the outer product of the top singular vector pair of $-\nabla_{\boldsymbol{Z}_1} f(\boldsymbol{X}^k, \boldsymbol{\eta}^k, \boldsymbol{Z}_1^{k-1}, \boldsymbol{Z}_2^{k-1})$, which can be computed efficiently in linear time using Lanczos algorithm [8]. Next we perform a line search to find $\theta_k = \operatorname{argmin}_{\theta \geqslant 0} h^1(\theta_k)$, where $h^1(\theta_k) = f(\boldsymbol{Z}_1^k) + \alpha \delta_k \theta_k$. Here $h^1(\theta_k)$ is the upper bound of the objective function at $\boldsymbol{Z}_1^k$, and one can compute $\theta_k$ efficiently in close form. Similarly, one can repeat the same procedure for computing $\boldsymbol{Z}_2^k$, and we use $h^2(\theta_k)$ to denote the linear search function for $\boldsymbol{Z}_2^k$.