[Reviews · NeurIPS 2016]

Reviewer 1

Summary

The paper introduces a model to predict the time of interactions between users and items, and the item which is the most promising for a user. This model builds upon a feature representation of users and items which shifts along time and is a weigthed sum of (i) past actions, (ii) past external features, and (iii) representations of users/items which did interact in the past. The law of time between interactions is represented by a Hawkes process based on the dot product between users and items representations. The model is slightly changed to allow for a loss function (negative log likelihood) convex in the parameters. Experiments show that the proposed model predicts the interaction time with a much better accuracy than previous similar model (LowRankHawkes) while it similarly predicts the item to interact with.

Qualitative Assessment

Modeling the evolution of tastes/properties of users/items along time is a challenging problem for Machine Learning. This paper proposes a solution to model that evolution in a continuous manner, without relying on cyclicity or constant shift assumptions. Experimental results support the proposed model. The paper was a pleasure to read. It puts apart technical points to focus on what is needed to understand the approach and use it. As an example, Section 4.2 recalls the conditional gradient algorithm but does not give closed forms of the gradients. Still, I was surprised/disappointed by three points: 1) Is matrix eta necessary ? This matrix contradicts the fact that a user/item evolve along time. Did you measure by any way the necessity for eta ? By example you could look at the Frobenius norm of eta compared to the Frobenius norm of $\Sum_ix_iX_i$, or indicates prediction results of the same model without eta. 2) Theorem 1 is not a Theorem. Theoretical results on nuclear norm minimization to recover rank minimization require assumptions on the matrices and on the observed entries. Such assumptions are not listed in current paper. The authors only give a sketch proof on how to support the proposed reparameterization. 3) Conclusions of Theorem 1 are excessive. Under some hypothesis, we probably can prove that the algorithm learns a matrix X of (low-)rank $k$, but that rank will be greater than the rank of A, B, C and D. Thereafter, X_1, X_5, X_6 and X_7 can live in different column spaces. To increase the practical usability of the proposed model, I'm looking forward extensions of that work (i) considering incremental update of the model to manage the never-ending flow of data, (ii) managing the exploration/exploitation tradeoff while using that model to recommend interactions to users. ****** typos ****** Figure 1(a): at the bottom, isn't it $I_j(t_k^***u***)$ as it feeds user $u$ representation (if so, correct also the caption). Figure 1(a): Arrows are a bit messy. I would expect the arrow from $U_u(t_1^u)$ to go to $I_i(t_l^u)$ as the effect of the interaction is taken into account in the future. Similarly, I would not expect any arrow from $q_j^{u,i}$ to $I_i(t_1^u)$; there is already an arrow from $q_j^{u,i}$ to $I_i(t_l^u)$. If these remarks make sens, there is also two arrows at time $t_l^u$ which should disappear. Figure 1(b): I would expect an arrow from node $j$ to represent the fact that the representation of that item is poorly aligned with the representation of user $u$. L.161: "[...] item $i$ at time ***$t$*** than [...]" L.162: "[...] item $i$ than item ***$j$***. [...]"

Confidence in this Review

2-Confident (read it all; understood it all reasonably well)


Reviewer 2

Summary

In this paper, the authors propose a novel recommendation algorithm that aims to model the co-evolutionary properties of users and items. Specifically, the input of problem are implicit feedback (instead of ratings in the Netflix competition), time and associated features. The main idea is to use three terms to model the time-dependent latent features of a user as shown in Eq.(1), and also use three terms to model the time-dependent latent features of an item as shown in Eq.(2). Empirical results on some synthetic and real data show the effectiveness of the proposed algorithm.

Qualitative Assessment

The idea of using three terms to model the latent features of a user or an item is interesting, while the novelty is a bit limited. Some comments for improvement: 1. The three real data sets are too dense, i.e., on average, an IPTV user is associated with 337 events, a Yelp user is associated with 194 events, and a Reddit user is associated with 5800 events. Are the data pre-processed in some way? The authors may include more details about the data processing and include results on sparse data. 2. The time complexity may be higher than the commonly used stochastic gradient descent method in recommender systems community. Some analysis and/or comparative empirical studies on this issue should be included. 3. Some important baselines that can model implicit feedback, time and features are missing, such as the factorization machine with pairwise loss [Steffen Rendle, ACM TIST 2012].

Confidence in this Review

2-Confident (read it all; understood it all reasonably well)


Reviewer 3

Summary

This paper proposed to extend previous work on dynamic user and item features in user-item interaction modeling to cover co-evolution between user and item features. A convex formulation of the parameter inference problem is proposed, and a resulting algorithm is described. Experimental results show substantial improvement over previous works.

Qualitative Assessment

Overall the paper seems to have proposed a method that performs very well on several real datasets. The proposed model also seems reasonable. I am not entirely convinced on the optimization algorithm, though: 1. The theorem does not seem to say anything more than a trivial fact. 2. Why is the 3rd term in (7) necessary? 3. There are no details on how to choose the parameters alpha, beta, and gamma. Plus, selecting all three parameters will largely increase the computational complexity. 4. Once X is estimated, there is no guarantee to get A,B,C,D out of it. Or is that not important at all? There is no justification.

Confidence in this Review

2-Confident (read it all; understood it all reasonably well)


Reviewer 4

Summary

The authors work on modelling the reoccurring interactions between users and items. They aim to both predict which interaction is the most probable to occur at any given time, and when will any given interaction most probably occur. They introduce a method that takes inspiration from temporal point processes and matrix factorization. The method is able to use content and collaborative information (but might not work in a purely collaborative filtering setting). Their approach seems to work well on the item recommendation problem, and to be a big improvement over previous methods for predicting the time of a given interaction.

Qualitative Assessment

Interesting work, unusual approach to recommendation systems. Good improvement over the previous LowRankHawkes paper. Unfortunately, the complexity of the method does not leave enough space for the experiments section. The lack of information makes it very hard to replicate the results. We need explanation about what are the base drift and the interaction features in the real datasets. How are the parameters tuned on the real datasets ? Cross-validation is only mentioned for the synthetic data. Are the results averaged over several runs for the stochastic models ? What does the error bars represent in the figures ? It is true that the format of the proceeding does not leave enough space for a thorough presentation, but you could have supplementary material and you could make the code available. I especially think that releasing your code is important, given the originality of your approach. Still in the experiments section, the competitors on the item prediction problem might not be the most appropriate. - TimeSVD++ is designed for rating prediction, not item recommendation. A method designed for item prediction, such as BPR-MF or EigenRec, would probably yield better results. - FIP is designed to combine information from a social network with collaborative filtering. What did you use as a social network in your experiments ? If no social network was available, FIP seems inadequate. One last remark: it is unclear to me whether your approach works in a purely collaborative filtering setting. It looks like you need to have at least base drift or interaction features for it to work. How would you do if you do not have those features ? Maybe you can set the interaction feature vector q to be a one hot encoding of the item/user. Whether that's the case or not, this aspect should be addressed more explicitly in the paper.

Confidence in this Review

2-Confident (read it all; understood it all reasonably well)


Reviewer 5

Summary

The authors propose co-evolutionary latent feature process for user-item interaction, and show that the model can be solved efficiently. In summary, although this work is not strong in modelling, it is good in terms of writing.

Qualitative Assessment

Time evolution features are promising research directions. Indeed, the recommendation should be time-dependent. The authors have a good motivation to conduct the research. Here are some detailed comments: 1. what is the formal definition of the function \kappa(.) in (1) and (2)? Please be specific. 2. The authors mentioned evolution of feature. Intuitively, such a assumption could be true. But I am wondering if there is any empirical support of it? And how could justify that the user and item features are "co-evolving"? 3. In Figure 4, we could see clearly there are changes of users' interests. But how could we know the interests are evolving?

Confidence in this Review

2-Confident (read it all; understood it all reasonably well)


Reviewer 6

Summary

This paper proposes a "coevolutionary" latent feature temporal point process model that captures coevolving user and item features over time. A convex optimization algorithm is presented for learning model parameters. An experimental evaluation on a synthetic dataset and several real-world datasets is used to compare the proposed model's predictive performance with competing approaches.

Qualitative Assessment

The proposed co-evolving latent feature process model is interesting, and appears to be an extension of the LowRankHawkes model referenced in this paper. The design and intuition behind the model is generally presented well, and a good case is made for the importance of considering mutual dependencies between evolving user and item latent features. The paper indicates that the model parameters can be learned efficiently using the proposed convex optimization algorithm, but some details regarding its efficiency are missing. For example, what is the computational complexity of the required gradient computations? Can the optimization algorithm be run stochastically on minibatches, to improve scalability when training on large datasets? While Figure 2c indicates that the optimization algorithm scales linearly with the number of events, how does it scale in terms of the number of users and items? Figure 1 is busy/complex, and does not help with understanding the proposed model. This figure should be improved and simplified so that it is easier to interpret. The experimental results show that the proposed model performs well compared to strong competing approaches on several datasets, particularly regarding time prediction. On item recommendation for the real-world datasets, however, the performance results compared to LowRankHawkes are mixed, with the proposed model being merely competitive with this baseline (or slightly worse than this baseline). Why is this the case?

Confidence in this Review

2-Confident (read it all; understood it all reasonably well)